



# Pollen observations at four EARLINET stations during the ACTRIS-COVID-19 campaign

Xiaoxia Shang[1], Holger Baars[2], Iwona S. Stachlewska[3], Ina Mattis[4], and Mika Komppula[1]

[1]Finnish Meteorological Institute, Kuopio, Finland
[2]Leibniz Institute for Tropospheric Research (TROPOS), Leipzig, Germany
[3]University of Warsaw, Faculty of Physics, Poland
[4]Deutscher Wetterdienst, Meteorologisches Observatorium Hohenpeißenberg, Hohenpeissenberg, Germany

*Correspondence to*: Xiaoxia Shang (xiaoxia.shang@fmi.fi)

**Abstract.** Lidar observations were analysed to characterize atmospheric pollen at four EARLINET (European Aerosol Research Lidar Network) stations (Hohenpeißenberg, Germany; Kuopio, Finland, Leipzig, Germany; and Warsaw, Poland) during the ACTRIS-COVID-19 campaign in May 2020. The re-analysis lidar data products, after the centralized and automatic data processing with the Single Calculus Chain (SCC), were used in this study, focusing on particle backscatter coefficients at 355 nm and 532 nm, and particle linear depolarization ratios (PDRs) at 532 nm. A novel method for the characterization of the pure pollen depolarization ratio was presented, based on the non-linear least square regression fitting using lidar-derived backscatter-related Ångström exponents (BAEs) and PDRs. Under the assumption that the BAE between 355 and 532 nm should be zero (± 0.5) for pure pollen, the pollen depolarization ratios were estimated: for Kuopio and Warsaw stations, the pollen depolarization ratios at 532 nm were of 0.24 (0.19–0.28) during the birch dominant pollen periods; whereas for Hohenpeißenberg and Leipzig stations, the pollen depolarization ratios of 0.21 (0.15–0.27) and 0.20 (0.15–0.25) were observed for periods of mixture of birch and grass pollen. The method was also applied for the aerosol classification, using two case examples from the campaign periods: the different pollen types (or pollen mixtures) were identified at Warsaw station, and dust and pollen were classified at Hohenpeißenberg station.

## 1 Introduction

Pollen is recognized as one of the major agents of allergy-related diseases, such as asthma, rhinitis, and atopic eczema (Bousquet et al., 2008). Gilles et al. (2020) state that pollen exposure weakens the immunity against some respirator viruses, e.g. corona virus, by diminishing the antiviral interferon response. As one important type of biogenic particles, pollen has various climatic and environmental impacts (IPCC, 2013). They can affect the solar radiation reaching Earth thus causing cooling effect; whereas their interactions with long-wave radiation warm the atmosphere. In addition, they can influence the cloud optical properties and cloud lifetime by acting as cloud condensation nuclei (Griffiths et al., 2012; Pope, 2010; Steiner et al., 2015) and ice nuclei (von Blohn et al., 2005; Diehl et al., 2001, 2002), thereby influencing climate. In favourable





conditions, pollen can be lifted into upper layers of the atmosphere and travel thousands of kilometres from source areas (Rousseau et al., 2008; Skjøth et al., 2007; Szczepanek et al., 2017).

In 2021, there are more than 1000 active pollen monitoring stations in the world (https://oteros.shinyapps.io/pollen_map/, last access: 1 Oct 2020; Buters et al., 2018). The majority of stations operate devices based on the Hirst principle (Hirst, 1952), e.g. Burkard pollen sample, using manual microscopy. Automatic pollen measuring devices are also available, having potential

for workload reduction and online pollen monitoring. These techniques are based on, e.g. image recognition such as Pollen Monitor BAA500 (Oteros et al., 2015), or fluorescence spectra such as Wideband Integrated Bioaerosol Sensor (WIBS) (Gabey et al., 2010; Savage et al., 2017) and Plair Rapid-E (Šauliene et al., 2019), or digital holography such as Swisens Poleno (Sauvageat et al., 2020), or light scattering such as pollen monitor KH-3000-01 (Miki and Kawashima, 2021). Nonetheless, those pollen detections are usually on the ground and/or roof level.

An increasing interest has arisen to investigate the vertical distribution of pollen in the atmosphere. Studies show that lidar measurements can detect the presence of pollen in the atmosphere, with a strong diurnal cycle on the pollen backscattering, and that the non-spherical pollen grains can generate strong depolarization of laser light (Bohlmann et al., 2019, 2021; Noh et al., 2013a, 2013b; Sassen, 2008; Sicard et al., 2016). Therefore, it is possible to observe pollen in the atmosphere using the depolarization ratio in the absence of other depolarizing non-spherical particles (e.g. dust). We have estimated the

depolarization ratio at 532 nm of atmospheric birch and pine pollen as 0.24 ±0.01 and 0.36 ±0.01 under certain assumptions using a recently developed algorithm based on a multi-wavelength Raman polarization lidar measurements (Shang et al., 2020). Using laser induced fluorescence (LIF) lidars, Saito et al. (2018) and Richardson et al. (2019) were able to detect the fluorescence spectrum of pollen in the atmosphere. Veselovskii et al. (2021) demonstrated that the presence of pollen in aerosol mixtures leads to an enhancement of the fluorescence backscattering which is beneficial to distinguish pollen from dust

particles. Aerosol classification schemes are available for both spaceborne lidar observations (Groß et al., 2015; Kim et al., 2018) and ground-based lidar networks (Baars et al., 2017; Nicolae et al., 2018). However, pollen (or biogenic aerosols in general) is not included, and is likely misclassified as dusty mixtures.

An intensive observation campaign, ACTRIS-COVID-19 campaign, was organized in May 2020, focusing on the lidar observations of aerosols during the relaxation period after the lockdown periods. Pollen presence was also identified by the

continuous lidar measurements at several stations, as spring is the typical pollen season. Study was conducted at four European lidar stations (Hohenpeißenberg, Germany; Kuopio, Finland, Leipzig, Germany; and Warsaw, Poland) for the pollen property retrieval. They were selected based on the availability of lidar products and the possible pollen presence from measurements or models for dust-free periods during the campaign. A novel simple method for the characterization of the pure pollen is proposed, based on the non-linear least square regression fitting, using lidar-measured vertical profiles of particle backscatter

coefficients at 355 nm and 532 nm, and particle linear depolarization ratios at 532 nm. It was applied to evaluate the pollen depolarization ratio at these stations. For two case examples in the campaign period it was also used for the aerosol classifications.



The paper is structured as follows. In Sect. 2, we introduce the campaign, stations, instrumentation, and proposed algorithm. In Sect. 3, the results of the pollen characterization and the aerosol classification are presented and discussed. The conclusions are given in Sect. 4.

## 2    Measurements, instrumentation, and methodology

### 2.1  Stations and campaign

The ACTRIS-COVID-19 NRT (near-real-time) lidar measurement campaign was performed between 1 to 31 May 2020, involving 21 stations of the European Aerosol Research Lidar Network (EARLINET, https://www.earlinet.org, last access: 1 Oct 2021). A map with the participating EARLINET stations can be found at EARLINET website (https://www.earlinet.org/index.php?id=covid-19, last access: 1 Oct 2021). This intensive observation campaign was organized within the ACTRIS (Aerosol, Clouds and Trace Gases Research Infrastructure, https://www.actris.eu, last access: 1 Oct 2021) initiative for studying the changes in the atmosphere during the COVID-19 lockdown and early relaxation period in Europe.

Based on the availability of the vertical profiles of backscatter coefficients at 355 and 532 nm and particle linear depolarization ratios at 532 nm for dust-free pollen periods during the campaign, four lidar stations (Hohenpeißenberg, Germany; Kuopio, Finland, Leipzig, Germany; and Warsaw, Poland; Table 1) were selected for the pollen investigation. These stations belong to the Raman and polarization lidar network PollyNET (Baars et al., 2016; http://polly.tropos.de, last access: 1 Oct 2021).

**Table 1. Information of EARLNET lidar stations involved in this study.**

| Station | ACTRIS code | Institute | Coordinates (lat, long, elevation a.s.l.) |
|---|---|---|---|
| Hohenpeißenberg | HPB | Deutscher Wetterdienst (DWD) Meteorological Observatory Hohenpeißenberg, Germany | 47.80°N, 11.01°E, 974 m |
| Kuopio | KUO | Finnish Meteorological Institute (FMI), Atmospheric Research Centre of Eastern Finland, Kuopio, Finland | 62.74°N, 27.54°E, 190 m |
| Leipzig | LEI | Leibniz Institute for Tropospheric Research, Leipzig, Germany | 51.35°N, 12.43°E, 125 m |
| Warsaw | WAW | University of Warsaw, Faculty of Physics, Poland | 52.21°N, 20.98°E, 112 m |

Hohenpeißenberg station (HPB) is situated on top of an isolated mountain in the foothills of the Alps at Hohenpeißenberg in Germany. The Observatory is a major Global Station of the Global Atmospheric Watch program. This rural site is surrounded by spruce forests (*Picea abies*) mixed with some common beeches (*Fagus sylvatica*), maple (*Acer platanoides*), and ash





(*Fraxinus*) trees. About a third of the area is pasture land. Kuopio station (KUO) is located ~ 18 km from the city centre of Kuopio, in Eastern Finland. This is a rural site mainly surrounded by forest. Dominant tree species include Silver birch (*Betula pendula*), Norway spruce (*Picea abies*) and Scots pine (*Pinus sylvestris*). Leipzig station (LEI) is located in in the lowlands of Eastern Germany. The surrounding is dominated by agricultural areas and some forest together with wetlands. Typical trees are birch, lime, beech, oak, maple, and pine among others. Main agricultural plants are all kinds of corn, maize, rape and grass.

The city of Leipzig itself has a lot of parks and a high biodiversity. Many kinds of trees and other plants can be found. The pollution level is medium to low, as Leipzig is usually well circulated by the dominant wind systems as no hills or mountains are around. Beside in times of intensive agricultural activity (early spring or late autumn) or periods of Saharan dust arrival, no depolarizing aerosol is observed in Leipzig, leading to a background particle depolarization ratio of ~ 0.01. Warsaw station (WAW) is located in the city centre of the capital of Poland, however, in nearby vicinity there are several green parks. In May,

typically observed pollen species are pine (*Pinus*), birch (*Betula*), and blue grass (*Poa*). The fungi spores represent very high contribution in vegetation season.

Birch pollen is recognised as one of the most important allergenic sources (D'Amato et al., 2007), which has a diameter around 20–30 µm and near-spherical shape with three pores on the edge. Beeches, maple and ash pollen is quite similar as birch pollen in terms of shape and size. Pine and spruce pollen grains, belonging to the *Pinaceae* family, are significantly larger, with the

diameter on the longest axis of ~65–80 µm or ~90–110 µm, respectively (Nilsson et al., 1977). They possess two air bladders which assist those pollen grains to be dispersed by wind despite their large size. The *Poaceae* family, known as grasses, comprises over 12 000 species classified into 771 grass genera (Soreng et al., 2015). Grass pollen grains are spheroidal to sub-oblate in shape with a single circular pore, whereas the size range is highly variable (García-Mozo, 2017; Joly et al., 2007; Salgado-Labouriau and Rinaldi, 2009). Microphotographs of pollen grains can be found at PalDat – a palynological database

(https://www.paldat.org, last access: 1 Oct 2021, Halbritter and Heigl, 2020).

## 2.2  Lidars and data processing

These four PollyNET stations are all equipped with ground-based multi-wavelength Raman polarization lidars Polly$^{XT}$ (Baars et al., 2016; Engelmann et al., 2016). Full details on the setup, principle of Polly$^{XT}$ can be found in Engelmann et al. (2016). Measurement capabilities of the lidars are somewhat different, but they all have emission wavelengths at both 355 and 532 nm,

and depolarization channels at 532 nm. The lidar near-real-time quick looks are publicly accessible at the PollyNET website (http://polly.tropos.de, last access: 1 Oct 2021).

Lidar data was processed in a centralized way using the Single Calculus Chain (SCC), with specific configurations and settings, and was made publicly available. The SCC is a tool for the automatic analysis of aerosol lidar measurements developed within EARLINET network (D'Amico et al., 2015, 2016; Mattis et al., 2016). The aerosol optical products after the re-analysis were

used; available on the THREDDS server (https://login.earlinet.org:8443/thredds/catalog/covid19re/catalog.html, last access: 1 Oct 2021). Out of all available data products, this study focused on particle backscatter coefficients (BSCs) at 355 nm and 532 nm, and particle linear depolarization ratios (PDRs) at 532 nm.



### 2.3 Ancillary data

In order to make sure that there is no dust contamination in the pollen properties retrieval, only dust-free periods were
considered in this study, which were identified using prediction of the NMMB/BSC-Dust (Non-hydrostatic Multiscale Model
/ Barcelona Supercomputing Center, Pérez et al., 2011; https://ess.bsc.es/bsc-dust-daily-forecast, last access: 1 Oct 2021). The
NMMB/BSC-Dust is an online multi-scale atmospheric dust model designed to accurately describe the dust cycle in the
atmosphere, and intended to provide short to medium-range dust forecasts for both regional and global domains. It provides
vertical profiles of dust concentration every 6 hours, with a horizontal resolution of 0.3°×0.3°. HYSPLIT (Hybrid Single-
Particle Lagrangian Integrated Trajectory, https://ready.arl.noaa.gov/HYSPLIT.php, last access: 1 Oct 2021) backward
trajectories were analysed to study the air mass origins.

Pollen types and concentrations were determined by the model forecasting and/or in situ measurements at the ground level
when available. The SILAM (System for Integrated modeLling of Atmospheric coMposition) dispersion model (Sofiev et al.,
2015; https://silam.fmi.fi, last access: 1 Oct 2021) provides the forecasts of pollen distribution over Europe, with 10 km and
1 h as spatial and time resolutions, respectively. Vertical profiles of pollen concentrations are available for 10 height levels
(with layer midpoint height from 12.5 m to 7725 m from the surface), including 6 pollen types (alder, birch, grass, mugwort,
olive, and ragweed pollen; Siljamo et al., 2013; Sofiev, 2017; Sofiev et al., 2013, 2015b). A Hirst-type Burkard pollen sampler
was placed 4 m above ground level (agl) at Kuopio station during the campaign to enable identification of pollen types and
concentration microscopically with a 2 h time resolution (more detailed descriptions can be found in Bohlmann et al., 2019
and reference therein). In Germany, the pollen monitoring is available online at 6 locations (including the Leipzig station),
using the fully-automatic Pollen Monitor BAA500 (Hund GmbH; https://www.hund.de/en/service/pollen-monitor, last access:
1 Oct 2021), that combines advanced computer aided microscopy, camera and image recognition technology to determine and
count pollen grains with a 3 h time resolution.

### 2.4 PDR vs BAE theory

Previous pollen studies show tendencies towards smaller Ångström exponents with increasing depolarization ratios (Bohlmann
et al., 2019, 2021; Shang et al., 2020), indicating the increasing impact of larger and non-spherical pollen particles. Here, we
investigate, mathematically, the relationship of the backscatter-related Ångström exponent (BAE) and the particle linear
depolarization ratio (PDR). Two aerosol populations, depolarizing ($d$) and non-depolarizing ($nd$) aerosols, are considered. The
total particle backscatter coefficient ($\beta_{total}$) is the sum of the backscatter coefficients of depolarizing ($\beta_d$) and non-depolarizing
($\beta_{nd}$) aerosols.

The BAE describes the wavelength dependence on the backscatter coefficients between two wavelengths $\lambda_1$ and $\lambda_2$ (Ångström,
1964):

$$BAE_x(\lambda_1, \lambda_2) = -\frac{\ln\left(\frac{\beta_x(\lambda_1)}{\beta_x(\lambda_2)}\right)}{\ln\left(\frac{\lambda_1}{\lambda_2}\right)} \qquad (1)$$



with the index $x$ for aerosol type, which can be $d$ (for *depolarizing* particle, e.g. *pollen*), $nd$ (for *non-depolarizing* particle, e.g.

*background*), or *total* (for total particles). The wavelength pair $(\lambda_1, \lambda_2)$ was selected as (355,532) in this study. For simplicity

of the later calculation, we introduce the parameter $\eta$:

$$\eta_x(\lambda_1, \lambda_2) = \left(\frac{\lambda_1}{\lambda_2}\right)^{-BAE_x(\lambda_1,\lambda_2)} . \tag{2}$$

From now on, the wavelength pair $(\lambda_1, \lambda_2)$ for $\eta$ and BAE expressions is omitted in the following derivations.

Shang et al. (2020) demonstrated the power-law relationship between the BAE of total particles ($BAE_{total}$) and the pollen

backscatter contribution (the ratio of the pollen backscatter coefficient and the total particle backscatter coefficient) (see Eqs.

4–5 in Shang et al., 2020). Similarly, the backscatter contribution of depolarizing or non-depolarizing aerosols can be expressed

as:

$$\begin{cases} \frac{\beta_d(\lambda_2)}{\beta_d(\lambda_2)+\beta_{nd}(\lambda_2)} = \frac{\eta_{total}-\eta_{nd}}{\eta_d-\eta_{nd}} \\ \frac{\beta_{nd}(\lambda_2)}{\beta_d(\lambda_2)+\beta_{nd}(\lambda_2)} = \frac{\eta_{total}-\eta_d}{\eta_{nd}-\eta_d} \end{cases} . \tag{3}$$

The particle linear depolarization ratio of the total particles ($\delta_{total}$), containing depolarizing and non-depolarizing aerosols,

can be calculated using the backscatter coefficients and the depolarization ratios of each type as:

$$\delta_{total} = \frac{\frac{\beta_d * \delta_d}{\delta_d+1} + \frac{\beta_{nd} * \delta_{nd}}{\delta_{nd}+1}}{\frac{\beta_d}{\delta_d+1} + \frac{\beta_{nd}}{\delta_{nd}+1}} . \tag{4}$$

We divide both numerator and denominator with the total particle backscatter coefficient, i.e. $(\beta_d + \beta_{nd})$, and replace the

expressions in Eq. (3). Simple conversion yields:

$$\delta_{total} = \frac{\eta_{total}(\delta_d-\delta_{nd})-(\eta_{nd}\delta_d\delta_{nd}+\eta_{nd}\delta_{nd}-\eta_d\delta_{nd}\delta_d-\eta_d\delta_{nd})}{\eta_{total}(\delta_{nd}-\delta_d)-(\eta_{nd}\delta_{nd}+\eta_{nd}-\eta_d\delta_d-\eta_d)} , \tag{5}$$

and after further rearrangements we obtain

$$\eta_{total} = \frac{\eta_{nd}\delta_d(\delta_{nd}+1)-\eta_d\delta_{nd}(\delta_d+1)+\eta_{nd}(\delta_{nd}+1)-\eta_d(\delta_d+1)}{(\delta_d-\delta_{nd})(\delta_{total}+1)} - \frac{\eta_{nd}(\delta_{nd}+1)-\eta_d(\delta_d+1)}{(\delta_d-\delta_{nd})} . \tag{6}$$

This equation can be expressed in a simplified way as:

$$\left(\frac{\lambda_1}{\lambda_2}\right)^{-BAE_{total}} = \frac{a_1+a_2}{(\delta_{total}(\lambda_2)+1)} - a_2 , \tag{7}$$

with two coefficients $(a_1, a_2)$ defined from four characteristic parameters $(\eta_{nd}, \eta_d, \delta_{nd}, \delta_d)$ as:

$$\begin{cases} a_1 = \frac{\eta_{nd}\delta_d(\delta_{nd}+1)-\eta_d\delta_{nd}(\delta_d+1)}{(\delta_d-\delta_{nd})} \\ a_2 = \frac{\eta_{nd}(\delta_{nd}+1)-\eta_d(\delta_d+1)}{(\delta_d-\delta_{nd})} \end{cases} . \tag{8}$$

The relationship between lidar-derived BAE and PDR of total particles is fixed for the mixture of two aerosol types. It can be

mathematically derived if the characteristic values of these two aerosol types ($BAE_d$, $\delta_d$, and $BAE_{nd}$, $\delta_{nd}$) are known. Synthetic

examples are given in Fig. 1 where the backscatter coefficients profiles of depolarizing ($BSC_d$), non-depolarizing ($BSC_{nd}$), and

total particles ($BSC_{total}$) were simulated. Under different initial characteristic values of depolarizing and non-depolarizing

particles, the PDR and BAE profile of total particles are different (e.g. Fig. 1b–c). The relationships between simulated BAE





and PDR under each assumption are shown in Fig. 1d: the bottom-right (top-left) boundary point is determined by the BAE and the depolarization ratio (DR) of the depolarizing (non-depolarizing) particles, shown as dark brown squares (light brown triangles); whereas the curve shape of fitting lines is determined by Eq. (7). Such a relationship is valid under two constraints: (i) only two aerosol populations present in the mixture, (ii) both DRs and BAEs of the two aerosol types should be different.

These two aerosol types can be dust and non-dust aerosols, or pollen and non-depolarizing background aerosols.

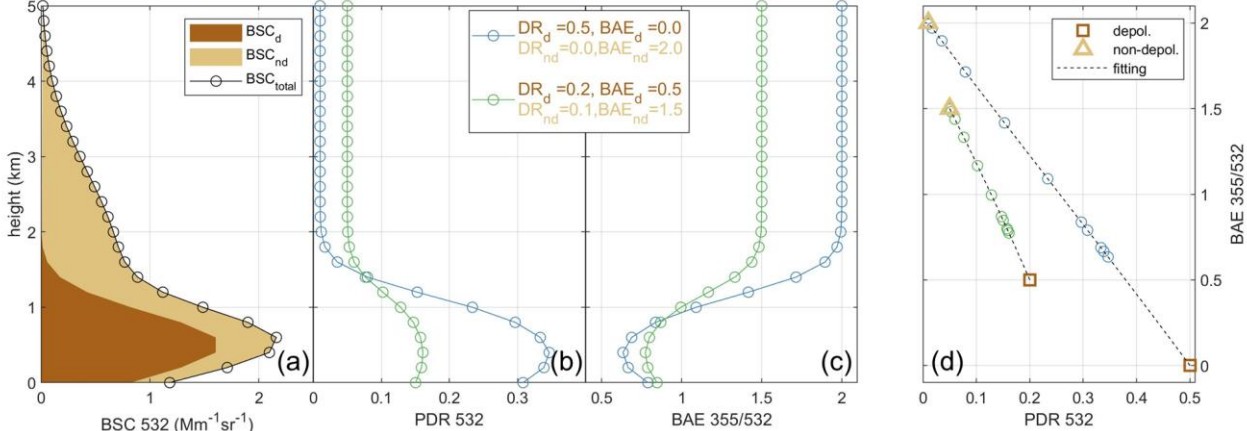

**Figure 1. (a) Synthetic vertical profile of the total particle backscatter coefficient (BSC$_{total}$); the shares of depolarizing (BSC$_d$), and non-depolarizing (BSC$_{nd}$) particles are given by dark and light brown area. Synthetic profiles of (b) the particle linear depolarization ratio (PDR) at 532 nm, and (c) the backscatter-related Ångström exponent (BAE) between 355 and 532 nm, under 2 group values of**
**the depolarization ratio (DR) and the BAE of depolarizing (*d*) and non-depolarizing (*nd*) particles (DR$_d$, DR$_{nd}$, BAE$_d$, BAE$_{nd}$). (d) Scatter plot of PDR and BAE for 2 groups of synthetic profiles, with dashed fitting lines and boundary points (dark brown squares and light brown triangles). Open circles present each bin.**

## 3  Results

### 3.1  Selected pollen periods

The pollen periods were selected for each station in May 2020 (Table 2), following the criterions: 1) dust-free as indicated by the NMMB/BSC-Dust model, 2) relatively high pollen concentrations (from the SILAM model forecasting and/or in situ measurements when available). Since the closest layer to the ground is assumed to contain the highest pollen concentration and share, the lowest layers were considered as the pollen layers in this study. In addition, the retrieved BSC at 532 nm and 355 nm should be larger than 0.05 and 0.1 Mm$^{-1}$ sr$^{-1}$, respectively.




**Table 2. Selected pollen periods for four stations. Source of possible dominant pollen types: a – SILAM model, b – Burkard pollen sampler, c – Pollen Monitor BAA500. Profile and bin numbers, layer heights, and lidar-derived optical values of selected layers for each station (mean values ± standard derivation) are given (PDR – particle linear depolarization ratio, BAE – backscatter-related Ångström exponent).**

| Station | Selected period in May 2020 (dd) | Possible dominant pollen types (source) | Profile (bin) number | Layer bottom (km agl) | Layer top (km agl) | PDR 532 | BAE 355/532 |
|---|---|---|---|---|---|---|---|
| KUO | 23–26 | Birch (a,b) | 9 (168) | $1.16 \pm 0.14$ | $2.21 \pm 0.32$ | $0.09 \pm 0.03$ | $1.52 \pm 0.42$ |
| WAW | 26–29 | Birch (a) | 20 (257) | $0.57 \pm 0.00$ | $1.28 \pm 0.32$ | $0.08 \pm 0.05$ | $1.31 \pm 0.45$ |
| HPB | 07–08 | Birch, grass (a) | 5 (39) | $0.71 \pm 0.03$ | $1.12 \pm 0.17$ | $0.04 \pm 0.01$ | $1.24 \pm 0.14$ |
| LEI | 26–27,30–31 | Birch, grass (a,c) | 4 (33) | $0.93 \pm 0.35$ | $1.36 \pm 0.42$ | $0.07 \pm 0.03$ | $1.10 \pm 0.30$ |


In Kuopio station, there was frequent rain in the first two thirds of May, and almost no pollen was measured by the Burkard sampler. Birch pollen was observed from 23 to 31 May, with the highest concentration of ~ 4000 $m^{-3}$ on 26 May. 23–26 May were selected as the pollen period, when there was clear sky. During the period, quite nice diurnal cycles (see Sect.3.3.1) were found from lidar observations with enhanced backscatter signals and volume depolarization ratios in the planetary boundary

layer.

In Warsaw station, two periods were selected in this study (see Sect.3.3.2): period no.1, birch pollen period from 26 to 29 May; period no.2, birch pollen mixture period on 31 May. High birch concentrations (with a median hourly value of 4800 $m^{-3}$ at the lowest level) were indicated from the SILAM model for both periods, with almost 0 concentration of the other 5 pollen types. In Sect. 3.2 only period no.1 is considered, whereas period no.2 will be discussed in Sect.3.3.2.

In Hohenpeißenberg station, high birch concentrations were found in 9 and 10 May with the highest value at the lowest level of ~ 180 $m^{-3}$, however there were dust presence (from NMMB/BSC-Dust) on these days. In order to avoid the dust mixture impact on the pollen property retrieval, two dust-free days (7 and 8 May, see Sect.3.3.3) were selected as the pollen period, where nice diurnal cycles of enhanced backscatter signals and volume depolarization ratios in the planetary boundary layer can be found. SILAM model forecasts suggest the presence of birch and a small amount of grass pollen, with the highest

concentration of ~ 60 $m^{-3}$ at the lowest level.

In Leipzig station, the number of available optical profiles was limited due to the frequent rain. From SILAM model, there were few occasions of pollen presence in May. Pollen period was selected as 4 days (26, 27, 30, 31), when there was mainly birch and grass pollen; only 4 lidar derived optical profiles of a full set were available in the period. The highest value of SILAM hourly pollen concentrations is about 100 $m^{-3}$. The Pollen Monitor BAA500 shows mean values of the daily pollen

concentration of 13 $m^{-3}$ and 26 $m^{-3}$ for birch and grass pollen during the period.





## 3.2 Characteristic values

Due to the small amount of profile numbers, values of all bins inside predefined pollen layers were used (see Table 2). The bottoms of the pollen layers are limited due to the overlap of the lidar instrument, whereas the tops are defined as the lowest observed layers based on the gradient method applied on both BSCs and PDRs. Averaged layer-mean values of PDRs in pollen

layers of four stations are slightly enhanced (from about 0.04 to 0.09), suggesting the presence of non-spherical particles in the atmosphere.

We assumed that inside the pollen layers there are only 2 aerosol types: pollen and non-depolarizing background aerosol ($bg$). Base on the approached presented in Sect. 2.4, we applied a simplified equation (similar to Eq. (7)) here:

$$y = \frac{a_1 + a_2}{(x+1)} - a_2 \qquad (9)$$

where $x$ is the bin value of measured PDR at 532 nm inside the pollen layer, i.e. $\delta_{\text{total}}(532)$; and $y$ is the bin value defined from BAE calculated by the measured BSCs at 355 and 532 nm inside the pollen layer, i.e. $y = \left(\frac{355}{532}\right)^{-BAE(355,532)}$. Note that if there are many profiles, it is possible to use the mean values of pollen layers instead of each bin.

The non-linear least square regression fitting, based on the Jacobian matrix, was applied using Eq. (9) to the dataset for each station to evaluate the coefficients ($a_1, a_2$), with values given in Fig. 2 with their standard deviations. The values of the

coefficients ($a_1, a_2$) are different for stations, as they are defined (Eq. (8)) from characteristic values of two aerosol types, i.e. pollen ($BAE_{\text{pollen}}, DR_{\text{pollen}}$) and non-depolarizing background aerosol ($BAE_{\text{bg}}, DR_{\text{bg}}$). For a good dataset, the unique solution can be found for the coefficients ($a_1, a_2$) with a high accuracy. But many solutions on the four characteristic values can result in the same coefficient couple ($a_1, a_2$), by reason of 2 equations with 4 unknows. Regarding the fitting Eq. (9), the value couple of $BAE_x$ and $DR_x$ of one pure particle type (*pollen* or *bg*) should be located on the fitting curve theoretically (or under

ideal conditions). Thus, with the knowledge of one parameter, the other can be evaluated. In reality, the depolarization ratio of the background particles ($DR_{\text{bg}}$) can be reasonably estimated or assumed, whereas the BAE of pure pollen ($BAE_{\text{pollen}}$) can be assumed to be 0, as pollen grains are quite large particles (e.g. birch pollen has a diameter around 20–30 µm). Hence, the other two characteristic parameters ($BAE_{\text{bg}}, DR_{\text{pollen}}$) can be calculated, and vice versa. Final estimations of characteristic parameters for all stations are given in Table 3. There are no values of the Ångström exponent for pure pollen in the literature;

for large particles such as dust, Mamouri and Ansmann (2014) reported extinction-related Ångström exponents between 440 and 675 nm, with values of -0.2 for coarse dust and 0.25 for total dust. If the true value of $BAE_{\text{pollen}}$ is assumed between -0.5 and 0.5, the possible ranges of $DR_{\text{pollen}}$ for each station can be given; refer to Table 3.

For Kuopio and Warsaw stations, the depolarization ratios at 532 nm of pure pollen (birch dominant) were found as 0.24, which is in agreement with the birch depolarization ratio of 0.24 reported in Shang et al. (2020) for lidar observations in Kuopio

in 2016. The pollen depolarization ratios at Hohenpeiβenberg and Leipzig stations have relatively smaller values, probably due to the mixture of birch and grass pollen, as indicated by SILAM model. Grass pollen, depending on the genera, can be more spherical in shape compared to birch pollen, thus smaller depolarization ratio is expected.



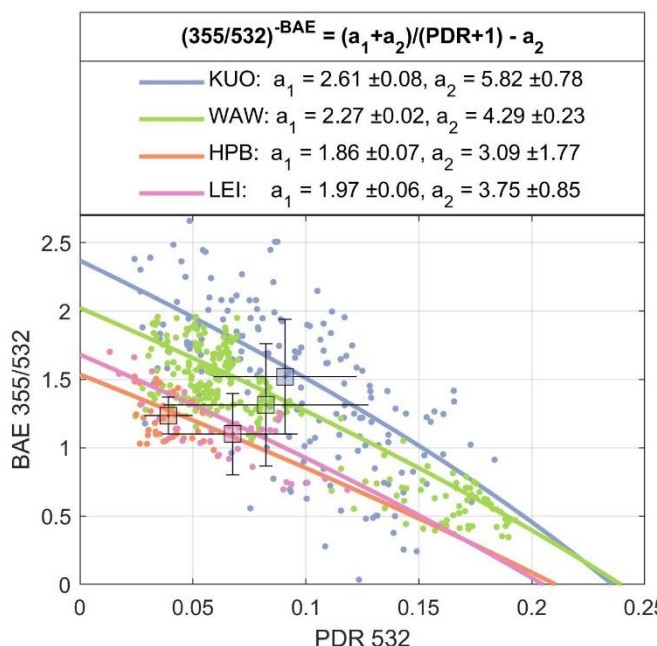

**Figure 2. Relationships of the particle linear depolarization ratio (PDR) at 532 nm and the backscatter-related Ångström exponent**
**(BAE) between 355 and 532 nm. All bins inside pollen layers are shown by dots for each station with different colours. Averaged layer-mean values are given by the square, with the variabilities shown by bars. Fitting regression lines (Eqs. (7) and (9)) are drawn with parameter values given in the legend.**

**Table 3. Characteristic values of background and pollen particles for pollen periods of four stations, derived from the regression fitting line in Fig. 2. DR: depolarization ratio at 532 nm. BAE: backscatter-related Ångström exponent between 355 and 532 nm.**
**"A" denotes the assumption.**

| Station | Background | | Pollen DR ($DR_{pollen}$) | |
| --- | --- | --- | --- | --- |
| | $DR_{bg}$ (A) | $BAE_{bg}$ | if $BAE_{pollen} = 0$ (A) | if $BAE_{pollen}$: 0.5 to -0.5 (A) |
| KUO | 0.03 | 2.1 | 0.24 | 0.20 to 0.27 |
| WAW | 0.02 | 1.9 | 0.24 | 0.19 to 0.28 |
| HPB | 0.01 | 1.5 | 0.21 | 0.15 to 0.27 |
| LEI | 0.01 | 1.6 | 0.20 | 0.15 to 0.25 |

### 3.3 Case examples

The present method (Sect. 2.4) was used to evaluate the characteristic values of the pure particle type, e.g. to estimate the pure pollen depolarization ratios, and a case example for Kuopio station is presented here (Sect. 3.3.1). It can also be applied for

the aerosol classification. Two case examples from the campaign periods are present (Sects. 3.3.2 and 3.3.3).





### 3.3.1    Kuopio – birch pollen

An overview of the selected pollen period at Kuopio station is given in Fig. 3. Bi-hourly concentrations from the Burkard sampler (Fig. 3a) at the roof level (~ 4 m agl) show birch pollen presence during the period, with other pollen types only accounted for ~ 2 %. The time–height plot of birch pollen concentrations from SILAM forecast is given in Fig. 3b, showing

that birch pollen can reach up to ~ 3 km agl with higher concentrations near ground. Polly$^{XT}$ lidar observations of the range-corrected signal (RCS) at 1064 nm and the volume depolarization ratio (VDR) at 532 nm are presented in Fig. 3c–d. A high aerosol load was observed within the first 3 km considering the strong backscatter signals. Enhance VDRs were correlated with higher birch concentrations, with diurnal cycles. A case example of lidar-derived optical profiles (time-averaged at 10:00–12:00 UTC on 26 May) is shown in Fig. 3e–f. The pollen backscatter contribution (the ratio of the pollen backscatter coefficient

and the total particle backscatter coefficient) at 532 nm was calculated based on the pollen depolarization ratio at 532 nm of 0.24 derived in Sect. 3.2. The layer mean value of the pollen backscatter contribution for the selected case is ~ 50 %. A clear tendency towards higher pollen contribution with increasing depolarization ratios and decreasing BAEs can be found, indicating the increasing impact of pollen in the aerosol mixture.

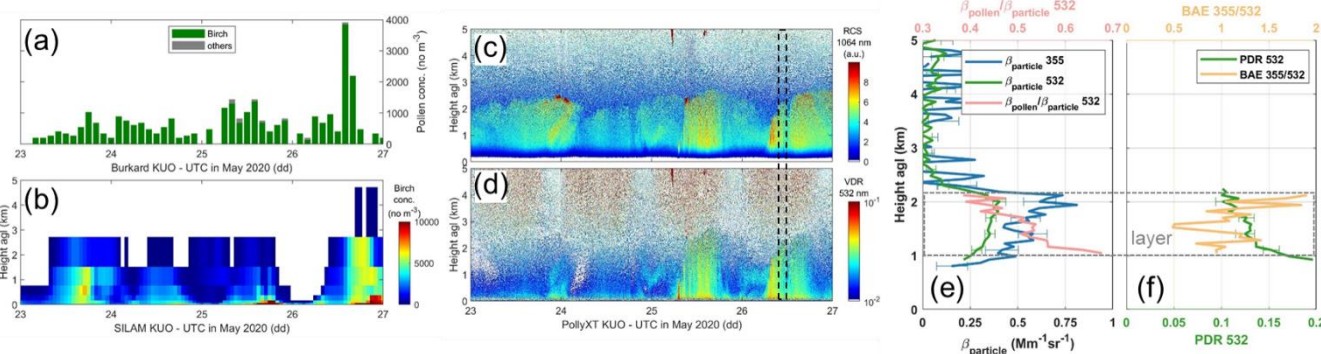

**Figure 3. Overview of the pollen period and a case example at Kuopio station. (a) Pollen concentrations from Burkard sampler at the roof level. (b) Birch pollen concentrations from SILAM model. Time–height cross section of (c) range-corrected signal (RCS) at 1064 nm and (d) Volume depolarization ratio (VDR) at 532 nm of Polly$^{XT}$. Vertical profiles of (e) backscatter coefficients, and the pollen backscatter contribution, (f) backscatter-related Ångström exponent (BAE) and the particle linear depolarization ratio (PDR), of the selected time period (black dashed box in c,d). Selected pollen layer is shown in grey dashed box.**

### 3.3.2    Warsaw – different pollen types

The time–height plot of VDRs at 532 nm from Polly$^{XT}$ at Warsaw station for 26–31 May 2020 is presented in Fig. 4a. Nice diurnal cycles of enhanced VDRs are visible, which are likely due to pollen presence in the atmosphere. The NMMB/BSC-Dust model suggests no dust presence below 7 km during the period. SILAM model, including 6 pollen types, forecasts that mainly birch pollen is present for the whole period. However, stronger VDR on 31 May was observed compared to previous

days. Two periods were defined (Table 4) for the comparison, separated by 30 May when low clouds and/or rain occurred. For the period no. 2, i.e. 31 May, only 2 profiles are available due to the low cloud. The non-linear least square regression fitting was applied using Eq. (9) to the dataset for two periods, separately, with results given in Fig. 4b and Table 4. The general





depolarization ratio of the background particles ($DR_{bg}$) at Warsaw station can be assumed as 0.02, the BAE of the background

particles were thus derived as quite closed values (1.9 or 1.8 for each period). Nevertheless, under the assumption of

$BAE_{pollen}= 0$, the pollen depolarization ratio of period no.2 was estimated as a higher value (0.29) than the one of period no.1

(0.24). The $DR_{pollen}$ value of period no.1 is in good agreement with the one of Kuopio station, for birch pollen. Higher $DR_{pollen}$

value of period no.2 suggests the additional presence of more non-spherical particles, e.g. pine pollen (Shang et al., 2020),

which are not included in SILAM model.

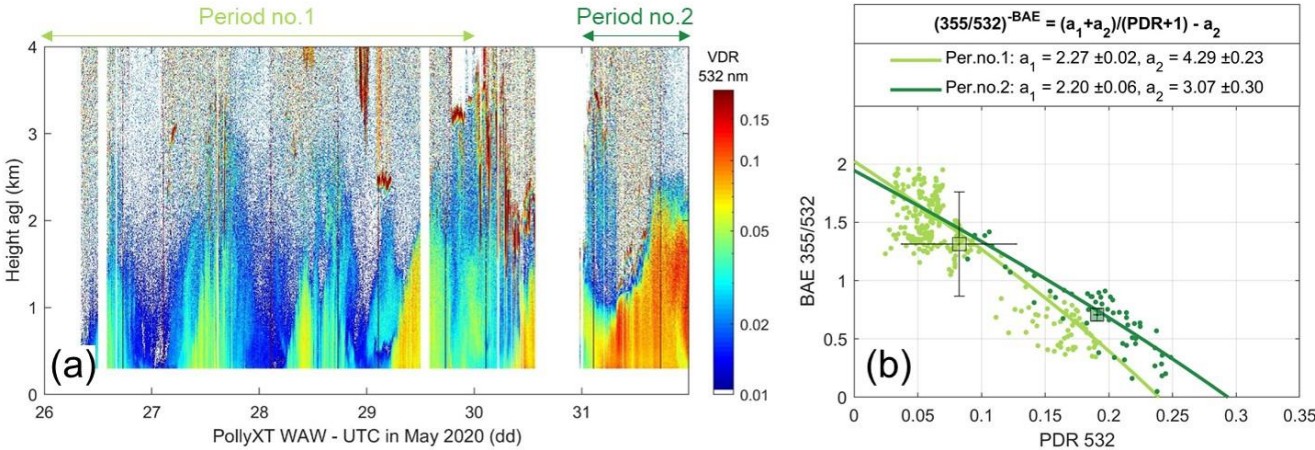

**Figure 4. (a) Time–height cross section of volume depolarization ratio (VDR) at 532 nm from Polly[XT] at Warsaw station on 26–31
May 2020. Selected two periods are indicated on the top. (b) Similar as Fig. 2, but for two periods at Warsaw station.**

**Table 4. Comparison of characteristic values of background and pollen / dust particles, for selected periods of Warsaw and
Hohenpeißenberg station. The index _d_ is used for the depolarizing particles (i.e. pollen or dust).**

| Station | Selected period in May 2020 (dd) | Profile (bin) number | Background $DR_{bg}$ | $BAE_{bg}$ | Possible depolarizing particle types | $DR_d$ if $BAE_d = 0$ | if $BAE_d$: 0.5 to -0.5 |
|---|---|---|---|---|---|---|---|
| WAW | Period no.1: 26–29 | 20 (257) | 0.02 | 1.9 | Pollen (birch) | 0.24 | 0.19 to 0.28 |
|  | Period no.2: 31 | 2 (56) | 0.02 | 1.8 | Pollen (birch mixture) | 0.29 | 0.23 to 0.35 |
| HPB | Period no.1: 07–08 | 5 (39) | 0.01 | 1.5 | Pollen (birch and grass) | 0.21 | 0.15 to 0.27 |
|  | Period no.2: 18 | 3 (19) | 0.01 | 1.7 | Dust | 0.32 | 0.24 to 0.40 |

### 3.3.3    Hohenpeißenberg – pollen and dust

Two periods were defined (Table 4) for the comparison study of pollen and dust particles observed in Hohenpeißenberg station.

In period no.1, only lowest layers were considered as pollen layers. A case example is given in Fig. 5, pollen presence can be

seen between 8:00 and 16:00 UTC close to the ground with enhanced backscatter signal and VDR. In period no.2, a lofted

aerosol layer with high VDRs, located at ~ 2 km at midnight and descending to ~ 1.5 km in the morning, was selected as the





dust layer (Fig. 6). The dust forecast at both Garmisch-Partenkirchen (47.47°N, 11.07°E) and Munich (48.15°N, 11.57°E)

stations (closest to Hohenpeißenberg station) of the NMMB/BSC-Dust model shows the dust layer at similar height. The air

mass sources, investigated by the backward trajectory analysis (HYSPLIT model), also shows that some of the particles were

transported from Sahara region.

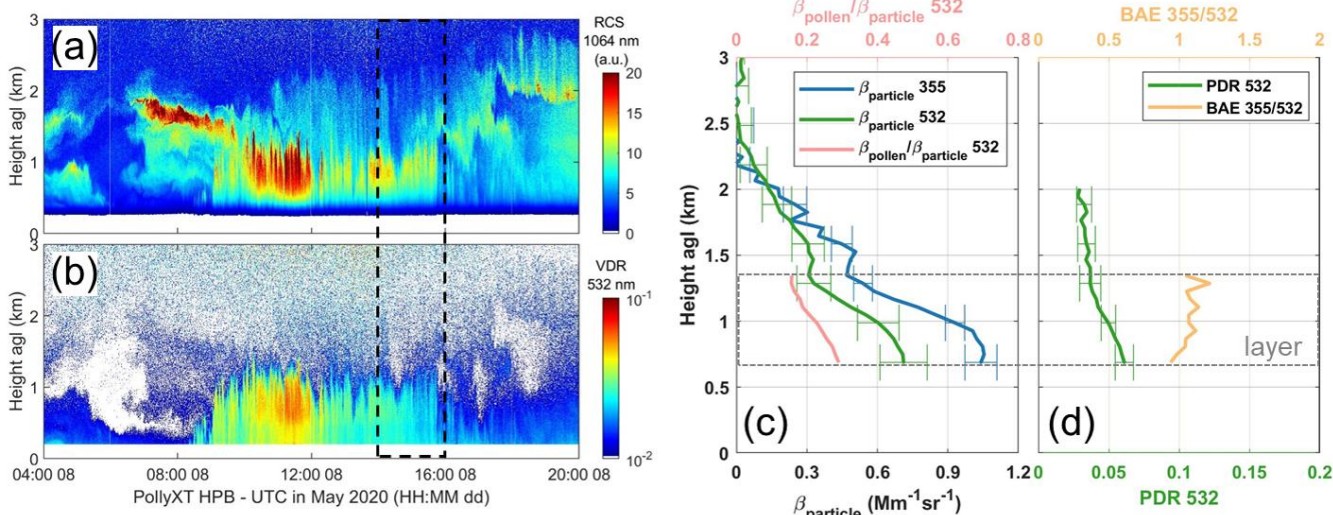

**Figure 5. Case example in period no.1 of Hohenpeißenberg station. Time–height cross section of (a) range-corrected signal (RCS) at**
**1064 nm and (b) volume depolarization ratio (VDR) at 532 nm from Polly$^{XT}$. Vertical profiles of (c) backscatter coefficients, and the**
**pollen backscatter contribution, (d) backscatter-related Ångström exponent (BAE) and the particle linear depolarization ratio**
**(PDR), of the selected time period (black dashed box in a,b). Selected layer is shown in grey dashed box.**

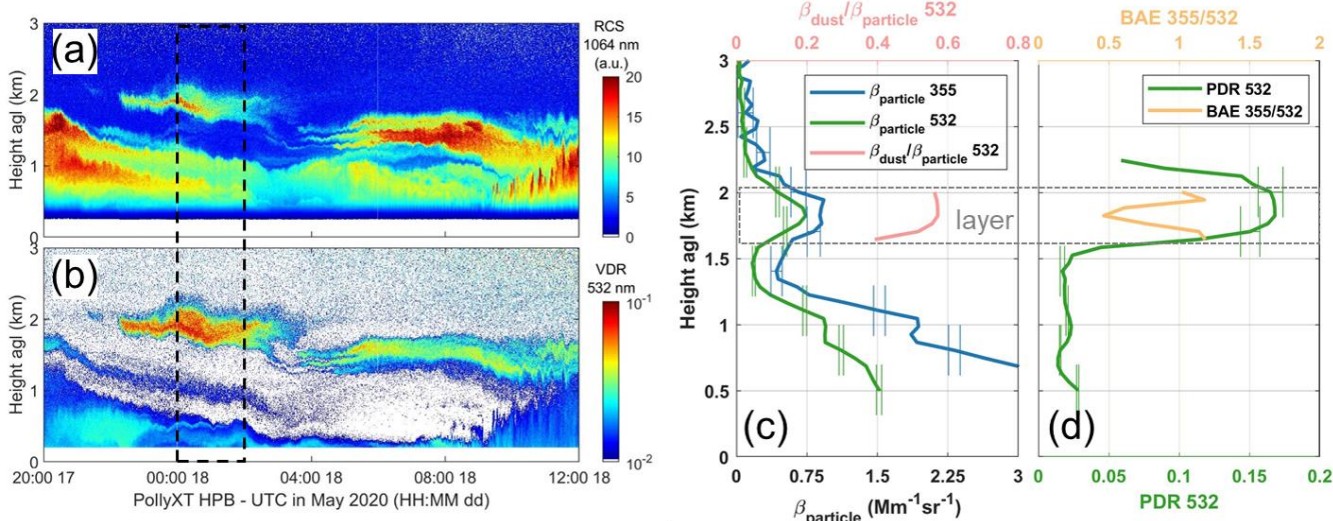

**Figure 6. Similar as Fig. 5, but in period no.2 of Hohenpeißenberg station.**

The non-linear least square regression fitting was applied using Eq. (9) to the dataset for two periods, with results given in Fig.

7 and Table 4. Even though the profile numbers are quite limited for both periods, the method was applied successfully using



all bins inside the selected layers. The depolarization ratio of the background particles ($DR_{bg}$) at Hohenpeißenberg station can

be assumed as 0.01, the BAE of the background particles were derived as 1.5 and 1.7 for two periods. Such a difference may

be due to the possible change on the background aerosol nature, as these two periods were separated by 10 days. If we assumed

that BAEs of both pollen and dust are equal to 0, the DRs of pollen and dust were estimated as 0.21 and 0.32, respectively.

Case examples of lidar-derived optical profiles are shown in Fig. 5c–d and Fig. 6c–d. The layer-mean backscatter contribution

of pollen (dust) for the selected case in period no.1 (no.2) was estimated as ~ 22 % (53 %), based on the evaluated pure

depolarization ratios of 0.21 (0.32) and BAE of 0. Using the presented method, the dust and pollen can clearly be classified

for this case study (e.g. Fig. 7). However, if the certain pollen type (e.g. pine pollen with 0.36 as $DR_{pollen}$ as reported in Shang

et al., 2020) has similar characteristic value as dust, the separation could be more challenging, and thus additional information

(e.g. the fluorescence as stated in Veselovskii et al., 2021) would be needed.

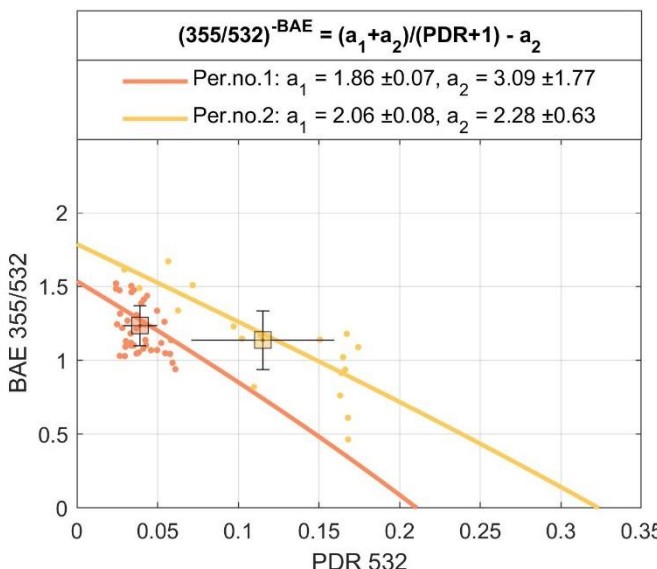

**Figure 7. Similar as Fig. 2, but for two periods at Hohenpeißenberg station.**

## 4    Summary and conclusions

During the ACTRIS-COVID-19 campaign in May 2020, continuous lidar measurements were performed at EARLINET

stations, with data (including optical property profiles) publicly available after the centralized and automatic data processing

with SCC. Four EARLINET and PollyNET lidar stations (Hohenpeißenberg, Germany; Kuopio, Finland, Leipzig, Germany;

and Warsaw, Poland) were selected for the pollen property retrieval during dust-free pollen periods, whereby we focused on

vertical profiles of particle backscatter coefficients at 355 and 532 nm, and particle linear depolarization ratios at 532 nm.

A novel method, based on the non-linear least square regression fitting using lidar-derived backscatter-related Ångström

exponent (BAE) and the particle linear depolarization ratio (PDR), was used for the characterization of the pure pollen

depolarization ratio. This easy-to-apply algorithm can estimate two coefficients to determine the relationship between PDR



and BAE. Such a relationship is valid under two constraints: (i) only two aerosol populations, depolarizing (e.g. pollen or dust)
and non-depolarizing (e.g. non-depolarizing background) aerosols, can be assumed in the aerosol mixture, (ii) both the
depolarization ratio (DR) and the BAE of the two aerosol types should be different. Mathematically (or under ideal conditions),
the PDR and BAE of a mixture of depolarizing and non-depolarizing aerosols, with whichever mixing rate, should follow the
derived relationship. Hence, with the knowledge of one parameter (PDR or BAE), the other can be evaluated. The characteristic
values of the pure aerosol type can be evaluated in this way, if one parameter is known or can be reasonably assumed.

Under the assumption that the BAE between 355 and 532 nm should be zero for pure pollen, the pollen depolarization ratios
were estimated: for Kuopio and Warsaw stations, the pollen depolarization ratios at 532 nm were found as 0.24 during the
birch dominant pollen periods; whereas for Hohenpeiβenberg and Leipzig stations, the pollen depolarization ratios were found
as 0.21 and 0.20 during the pollen period when there was a mixture of birch and grass pollen. However, the uncertainty on the
assumed BAE of pure pollen will introduce non-negligible bias. If the true value of pollen BAE is between -0.5 and 0.5, relative
uncertainties on estimated pollen depolarization ratios were found between 14–30 %. The present method was also applied for
the aerosol classification, using two case examples from the campaign periods. The different pollen types (or pollen mixtures)
were identified at Warsaw station, and dust and pollen were classified at Hohenpeißenberg station.

This study shows that automatically retrieved lidar data profiles (using SCC) are suitable for pollen characterizations. The
proposed methodology demonstrated a first step towards automated pollen detection in lidar networks.


*Data availability*. ACTRIS Aerosol Remote Sensing COVID-19 campaign data of May 2020:
https://doi.org/10.21336/gen.xmbc-tj86. Re-analysis aerosol optical products are available on the THREDDS server:
https://login.earlinet.org:8443/thredds/catalog/covid19re/catalog.html, last access: 1 Oct 2021. Optical products used in this
manuscript: DOI: http://doi.org/10.23728/fmi-b2share.959be96f095640578eb5a7dc335c8b46.


*Author contributions*. XS analysed the data, developed the algorithm, and wrote the manuscript. HB, ISS, IM, MK are the
principal investigator (PI) of the LEI, WAW, HPB, KUO stations, respectively. All authors ensured the high-quality operation
of the respective lidars. All authors reviewed and commented on the manuscript.

*Competing interests*. The authors declare that they have no conflict of interest.

*Special issue statement*. This article is part of the special issue "Quantifying the impacts of stay-at-home policies on
atmospheric composition and properties of aerosol and clouds over the European regions using ACTRIS related observations".
It is not associated with a conference.


*Acknowledgements*. The authors acknowledge the data and/or images from the NMMB/BSC-Dust model, operated by the
Barcelona Supercomputing Center (http://www.bsc.es/ess/bsc-dust-daily-forecast, last access: 1 Oct 2021). The authors



gratefully acknowledge the NOAA Air Resources Laboratory (ARL) for the provision of the HYSPLIT transport and dispersion model and/or READY website (https://www.ready.noaa.gov, last access: 1 Oct 2021) used in this publication. The authors acknowledge the Biodiversity Unit of University of Turku, the core personnel Annika Saarto and Sanna Pätsi, for the analysis of the pollen samples at Kuopio station. The authors acknowledge the pollen data of the Pollen Monitor BAA500 (Hund GmbH; https://www.hund.de/en/service/pollen-monitor, last access: 1 Oct 2021). The authors acknowledge the SILAM team, especially Mikhail Sofiev and Rostislav Kouznetsov, for the provision of SILAM model. The authors thank Simo Heikkinen for his help on the regression fitting algorithm. Warsaw lidar station measurements and data evaluation are performed in a team effort; during May 2020 the core personnel involved was I.S.Stachlewska, D.Szczepanik and R.Fortuna. Leipzig stations appreciates the contributions of all the individuals that have been involved in supporting, enabling, and maintaining Polly measurements and the Pollen observations. The research leading to the SCC results is supported by the European Commission under the Horizon 2020 - Research and Innovation Framework Programme, H2020-INFRADEV-2019-2, Grant Agreement number: 871115.

*Financial support.* This research has been supported by the Academy of Finland (projects no. 310312 and 329216). EARLINET stations acknowledge the support of ACTRIS; ACTRIS has received funding from the European Union's Horizon 2020 research and innovation programme under grant agreement numbers: 654109 (ACTRIS-2), 759530 (ACTRIS-PPP), 871115 (ACTRIS-IMP), 824068 (ENVRI-FAIR). The SCC development has been funded by the ACTRIS Research Infrastructure Project by the European Union's Horizon 2020 research and innovation programme under grant agreement no. 654109 and previously under the grant no. 262254 in the 7th Framework Programme (FP7/2007–2013). Warsaw station (UW) acknowledges support of European Space Agency, POLIMOS-4000119961/16/NL/FF/mg.

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
