# Peer review of "Pollen observations at four EARLINET stations during the ACTRIS-COVID-19 campaign"

_Atmospheric Chemistry and Physics, 2021_

## Author Comment (AC1)

**Response to Referee #1**

Thank you for carefully reading the manuscript and providing useful suggestions to improve the paper. The replies to the referee comments are given below. The referee comments are in blue with our responses in black. The sentences in the manuscript are between the quotation marks, with the modifications in the revised manuscript in red.

This paper presents a method to classify pollen types and determine the degree of mixing with other aerosols using the lidar-derived depolarization ratio and the backscattering coefficient at two wavelengths. Although there is a prerequisite that there should be no dust particles that can increase the depolarization ratio, it is judged that it is meaningful in that it is presented as a method to understand the distribution, type, and mixing degree of pollen in the atmosphere. Therefore, it is judged that this paper can be published in the relevant journal. However, it would be better if the following contents were corrected or added before posting.

Minor Comments

Line 53 : "ACTRIS-COVID-19" It is explained in section 2.1, but I would like the explanation of the abbreviation to come first.

Thank you for the comments. We made modifications in the revised version:
In the abstract:
"

Lidar observations were analysed to characterize atmospheric pollen at four EARLINET (European Aerosol Research Lidar Network) stations (Hohenpeißenberg, Germany; Kuopio, Finland, Leipzig, Germany; and Warsaw, Poland) during the ACTRIS (Aerosol, Clouds and Trace Gases Research Infrastructure) COVID-19 campaign in May 2020.
"
In section 1:
"

An intensive observation campaign, ACTRIS-COVID-19 campaign, was organized in May 2020, within the ACTRIS (Aerosol, Clouds and Trace Gases Research Infrastructure, https://www.actris.eu, last access: 1 Oct 2021) initiative for studying the changes in the atmosphere during the COVID-19 lockdown and early relaxation period in Europe.
"
In section 2.1:
"

The ACTRIS-COVID-19 NRT (near-real-time) lidar measurement campaign was performed between 1 to 31 May 2020, involving 21 stations of the European Aerosol Research Lidar Network (EARLINET, https://www.earlinet.org, last access: 1 Oct 2021). A map with the participating EARLINET stations can be found at EARLINET website (https://www.earlinet.org/index.php?id=covid-19, last access: 1 Oct 2021). This intensive observation campaign was focused on the lidar observations of aerosols during the relaxation period after the lockdown periods.
"

Line 193 : "the retrieved BSC at 532 nm and 355 nm should be larger than 0.05 and 0.1 Mm-1 sr-1, respectively." It's had better to explain why you use those two values.

Thank you for the comments. We have added descriptions in the revised version:
"

In addition, the retrieved BSC at 532 nm and 355 nm should be larger than 0.05 and 0.1 $Mm^{-1}$ $sr^{-1}$, respectively. These threshold values were adapted from the ones used in Baars et al. (2017), in which quasi-BSC at 1064 nm below 0.01 $Mm^{-1}$ $sr^{-1}$ or 0.2 $Mm^{-1}$ $sr^{-1}$ was classified as "Clean atmosphere" or "Non-typed particles/low concentration", respectively.
"

Line 223: "the overlap of the lidar instrument," It would be nice to indicate how far the affected altitude is.

Thank you for the comments. We have added descriptions in the revised version:
"

The bottoms of the pollen layers are limited due to the overlap of the lidar instrument (the lowest reliable height after the quality control tests is about 900, 500, 700, or 600 m agl, for KUO, WAW, HPB, or LEI, respectively), whereas the tops are defined as the lowest observed layers based on the gradient method applied on both BSCs and PDRs.
"

Line 311 : "NMMB/BSC-Dust model" The text keeps referring to the results of the model, but not showing the model results. How about showing the model results for the time period used in the study as an appendix?

Thank you for the suggestion. We will add a supplement (2. Dust concentration profiles from the NMMB/BSC-Dust model) to show the model results.

In section 3.1:
"

The pollen periods were selected for each station in May 2020 (Table 2), following the criterions: 1) dust-free as indicated by the NMMB/BSC-Dust model (see Supplement), 2) relatively high pollen concentrations (from the SILAM model forecasting and/or in situ measurements when available).

"

In section 3.3.3:
"

The dust forecast at both Garmisch-Partenkirchen (47.47°N, 11.07°E) and Munich (48.15°N, 11.57°E) stations (closest to Hohenpeißenberg station) of the NMMB/BSC-Dust model shows the dust layer at similar height (see Supplement).

"

**Response to Referee #2**

Thank you for carefully reading the manuscript and providing useful suggestions to improve the paper. The replies to the referee comments are given below. The referee comments are in blue with our responses in black. The sentences in the manuscript are between the quotation marks, with the modifications in the revised manuscript in red.

The paper reports a study on the discrimination of pollen based on observations from four lidar of similar setup, performance and data processing. The lidar observed pollen-laden atmospheres and provided colour and depolarization ratios. The presence of pollen, and the absence of dust which would hamper the detection and discrimination methodology, have been confirmed by models from dust and pollen distribution over europe and, when and where available, by direct in situ pollen detection and characterization. The study continues and extends previous works by some of the same authors. It is interesting and certainly deserves to be published after a suitable revision, also in view of the fact that the proposed method finds application not only to discriminate different pollen but whenever external mixtures of two aerosol are present, under the condition that both DRs and BAEs of the two aerosol types should be different.

The conditions under which the method is applicable are rather stringent and I don't know how frequently they can be encountered in reality. This is probably the main flaw of the article, not having discussed how extensible the method and its results are in general, beyond the single case studies presented here, which essentially validate the products of two models on dust transport and presence of pollen, which are initially used to select case studies.

Thank you for the comment, we have added a discussion of the method application to 3 aerosol type mixture using synthetic examples.
In section 2.4:
"

Such a relationship is valid under two constraints: (i) only two aerosol populations present in the mixture, (ii) both DRs and BAEs of the two aerosol types should be different. These two aerosol types can be dust and non-dust aerosols, or pollen and non-depolarizing background aerosols. The method application for synthetic examples of three aerosol types in the mixture is present and discussed in the Supplement. For two (or more) types of depolarizing aerosols and one non-depolarizing aerosol mixture, the estimated $DR_d$ represent a combination of two (or more) depolarizing aerosols, with a value between the characteristic (pure) DRs of each type. However, authors recommend using the method under the constraints mentioned above.

"
We also added one section in the Supplement:

**1. Synthetic examples for 3 aerosol mixture**

The method presented in the manuscript can be applied if there are only two aerosol populations present in the mixture. Here we discussed two cases when there are 3 aerosol types present in the mixture, using synthetic examples. In these cases, we assume there are dust, pollen and non-depolarizing background aerosols in the aerosol mixture and chosen the input characteristic values shown in Table S1. The backscatter coefficients profiles (at 532 nm) of each type were simulated (Fig. S1–S2 a), as well as the total particle backscatter coefficient. Using these initial characteristic values, the particle linear depolarization ratio (PDR) at 532 nm and the backscatter-related Ångström exponent (BAE) between 355 and 532 nm of total particles were calculated (Fig. S1–S2 b–c). The scatter plot of all bins (black circles) in the simulated profiles is given in Fig. S1–S2 d.

**Table S1. Initial characteristic values of dust, pollen and non-depolarizing background aerosols, as input of the synthetic examples.**

| Aerosol type | Depolarization ratio at 532 nm ($\delta$) | Backscatter-related Ångström exponent 355-532 (BAE) | Half width (Gauss.) | Layer centre case 1 (Fig. S1) | Layer centre case 2 (Fig. S2) |
|---|---|---|---|---|---|
| Dust | 0.40 | 0 | 1 km | 2.5 km | 1.5 km |
| Pollen | 0.30 | 0 | 1 km | 1.5 km | 1.5 km |
| Background | 0.01 | 2 | 3 km | 2.0 km | 2.0 km |

[Figure]

**Figure S1. Case 1: (a) Synthetic vertical profiles of the backscatter coefficient (BSC) at 532 nm of dust, pollen, background and total particles. Synthetic profiles of (b) the particle linear depolarization ratio (PDR) at 532 nm, and (c) the backscatter-related Ångström exponent (BAE) between 355 and 532 nm, of total particles, under initial values given in Table S1 – case 1. Two layers are defined (layer 1 – orange, layer 2 – green) in dashed boxes (a–c). (d) Scatter plot of PDR and BAE for the synthetic case. The dashed fitting lines are determined by Eq. (7), for layer 1 (orange), layer 2 (green), and all bins (black). Colour dots (inside each layer) and black circles (entire profile) present each bin.**

**Case 1:**

We defined 2 layers here (orange and green dashed boxes in Fig. S1 a–c): layer 1, from 3.0 to 3.8 km, where there are only dust and background aerosols present; layer 2, from 0.2 to 1.0 km, where there are only pollen and background aerosols present. As in both layers, there are only two aerosol types (one depolarizing and the other non-depolarizing), perfect fitting lines can be found by applying the method, i.e. Eq. (7) in the manuscript (orange and green dashed lines in Fig. S1 d). However, bins outside these 2 layers (black circles without colour) do not fit to either fitting line, but locating between them or on the top-left edge (where there are only background aerosols). When applying the method to all bins, a fitting line (in dashed black) was found between the ones of layer 1 and layer 2. As a result, under the assumptions that there are two aerosol types in the given aerosol mixture and that the BAE of this "depolarizing aerosol" should be zero, the pure depolarization ratio of this "depolarizing aerosol" was estimated as 0.36, which is a value between the initial depolarization ratios of pure dust (0.4) and pure pollen (0.3).

[Figure]

**Figure S2. Similar as Fig. S1, but for Case 2 (Table S1).**

**Case 2:**

In this case, the initial layer heights and layer depths of dust and pollen aerosol were selected to be the same (Table S1, Fig. S2 a). As these two depolarizing aerosols are well mixed with homogenous mixing ratio in this case, a perfect fitting line can be found for the PDR and BAE of total particles (black dashed line in Fig. S2 d) with all bins located on the fitting line, resulting a characteristic depolarization ratio of 0.36 under the assumption of characteristic BAE of zero. The theoretical fitting lines (from equations using initial values of Table S1) for dust and background aerosol mixture or pollen and background aerosol mixture are also given as orange or green dashed lines.

Not that the cases in the reality can be more complicated than the considered examples.

For the given examples, we can conclude that for an aerosol mixture of two types of depolarizing aerosols and one non-depolarizing aerosol, the estimated depolarization ratio for "depolarizing aerosols" represent the combination of two present depolarizing aerosols, with a value between the characteristic (pure) depolarization ratios of each type. For aerosol mixture with more types, the estimated depolarization ratio for "depolarizing aerosols" represent a combination of all present depolarizing aerosols, with a value inside the range of the characteristic (pure) depolarization ratio values of each aerosol type.

However, authors recommend using the method under the constraints mentioned in the manuscript.

More specifically, how good is the hypothesis of a characteristic depolarization and angstrom coefficient of a given pollen species, independent of the pollen aging phenomena and of the current relative humidity conditions at the time of observation, given the natural hygroscopicity of the pollen?

I encourage the authors to better explore the effect this crucial aspect on their hypotheses and results, exploring the dependence of the results on RH at the time of observation and perhaps also - if deducible - on the time elapsed since pollen was released into the atmosphere.

Thank you for the comment.

There were studies showing that relative humidity (RH) can affect the size and shape of pollen grains, which leads to different optical properties. At humid conditions, pollen grains swell by taking up water internally and after reaching a relative humidity over 89% external wetting of the pollen surface can occur (Griffiths et al., 2012) and thus could affect their shape at extreme humidities (e.g. >89%). In our previous pollen study (Bohlmann et al. 2019), we have investigated the relationship between the lidar derived optical properties with the lidar measured relative humidity in the atmosphere condition in Kuopio site, but didn't find any correlation (in RHs between 40-65%).

We have checked the vertical profiles of relative humidity from the CAMS reanalysis data (Inness et al. 2019, http://www.atmos-chem-phys.net/19/3515/2019/) for the time periods of each station, the RH at the low altitudes ranged between 20% and 80%, showing that our measurements were not affected by extreme humidity events and represent values for pollen under ambient atmospheric conditions in the spring season. Nevertheless, the measurements can be affected by extreme humidity event. And we have added such discussions in the revised version, at the end of section 3.2:

"

These measurements were not affected by extreme meteorological events and represent values for pollen under ambient atmospheric condition in the spring season (similar conclusions in Bohlmann et al. 2019). Note that different characteristic values of pollen could be observed under extreme humid or extreme dry conditions: i) pollen grains can be fold up and change the shape while dehydrating, e.g. commercially available pollen for laboratory measurements; ii) pollen grains can swell by taking up water especially after reaching a relative humidity over 89 % (see fig.2 in Griffiths et al., 2012).

"

The pollen observed are in general mostly local pollen, and we only consider the measurement for the layer closest to the ground (boundary layer aerosol) in this manuscript. We don't have means to evaluate the time elapsed, but this could be an interesting investigation, e.g. by models.

I believe, on the contrary, a variability of the optical characteristics of pollen is probable, depending on the present state and their specific history in the atmosphere. Likewise, apart from some particular stations (the authors cite Leipzig among them) it is possible that also the optical characteristics of the background aerosol are not constant to the degree required by the authors to apply their method. Given these drawbacks how the results here reported can be generalized?

Indeed, the characteristic values of background aerosol or of pollen can be estimated separately. Please see a more detailed reply for this and further referee's comment below for the line number 240.

Some more specific point the authors may wish to make clearer are the following:

line 172 until the end of the paragraph: I found the description of the simulation results quite confusing, especially panel d in figure 1, and the caption doesn't help. In particular, it is stated in the text that the dashed lines in panel b - which appear to be straight - descend from Eq. (7) - which does not look like a straight line.

We have made some modifications in the revised version to make it clearer. We have also changed the figure legend and figure caption as flowing for the clarity.
"

The relationship between lidar-derived BAE and PDR of total particles is fixed for the mixture of two aerosol types. It can be mathematically derived if the characteristic values of these two aerosol types ($BAE_d$, $\delta_d$, and $BAE_{nd}$, $\delta_{nd}$) are known. Synthetic examples are given in Fig. 1 where the backscatter coefficients profiles of depolarizing ($BSC_d$), non-depolarizing ($BSC_{nd}$), and total particles ($BSC_{total}$) were simulated. Under different initial characteristic values (case 1 or case 2) of depolarizing and non-depolarizing particles, the PDR and BAE profile of total particles are different (e.g. Fig. 1b–c in blue or green). The relationships between simulated BAE and PDR under each assumption are shown in Fig. 1d: the bottom-right (top-left) boundary point is determined by the BAE and the depolarization ratio (DR) of the depolarizing (non-depolarizing) particles, shown as dark brown squares (light brown triangles); whereas the curve shape of fitting lines is determined by Eq. (7), i.e. different values of $a_1$ and $a_2$ defined by Eq. (8). Note that the two boundary points are independent, as they are determined separately by the characteristic values (BAE and DR) of each aerosol type. Such a relationship is valid under two constraints: (i) only two aerosol populations present in the mixture, (ii) both DRs and BAEs of the two aerosol types should be different. These two aerosol types can be dust and non-dust aerosols, or pollen and non-depolarizing background aerosols.

[Figure]

**Figure 1. (a) Synthetic vertical profile of the total particle backscatter coefficient (BSC$_{total}$) at 532 nm; the shares of depolarizing (BSC$_d$), and non-depolarizing (BSC$_{nd}$) particles are given by dark and light brown area. Synthetic profiles of (b) the particle linear depolarization ratio (PDR) at 532 nm, and (c) the backscatter-related Ångström exponent (BAE) between 355 and 532 nm, under 2 group of initial values (case 1 in blue, and case 2 in green) of the depolarization ratio (DR) and the BAE of depolarizing ($d$) and non-depolarizing ($nd$) particles (DR$_d$, DR$_{nd}$, BAE$_d$, BAE$_{nd}$; values given in the legend). (d) Scatter plot of PDR and BAE for 2 synthetic cases. The dashed fitting line of each case is determined by Eq. (7) with parameters ($a_1$ and $a_2$) given. The boundary points (dark brown squares and light brown triangles) are defined by the initial values (shown in the legend in b-c). Open circles present each bin.**

"

The dashed lines in panel (d) are not straight; it follows the Eq.7: $\left(\frac{\lambda_1}{\lambda_2}\right)^{-BAE_{total}} = \frac{a_1+a_2}{(\delta_{total}(\lambda_2)+1)} - a_2$, where $BAE_{total}$ is shown in y-axis and $\delta_{total}$ is shown in x-axis. For the case 2 of Fig.1, we can see the slight difference between the fitting line and the straight line (in black) in the following figure:

[Figure]

As mentioned in the manuscript, the curve depends on the initial values, here we give another example of 2 cases, which shows bigger curve:

[Figure]

In table 2 they talk about "bins" without having defined them either in spatial or temporal terms. Moreover, it is not clear how the averages of the optical parameters shown in the table were calculated.

Thank you for point it out.
We added descriptions in section 2.2 for the spatial and temporal resolution of used optical profiles.
"

Out of all available data products, this study focused on particle backscatter coefficients (BSCs) at 355 nm and 532 nm, and particle linear depolarization ratios (PDRs) at 532 nm. The processing vertical resolution is ~ 60 m, and the integration time is of 2 h or less (depending on the cloud free time available).
"

We have also added descriptions in the table 2 caption and in section 3.2 for the clarity about the averaged values:
"

**Table 2. Selected pollen periods for four stations. Source of possible dominant pollen types: a – SILAM model, b – Burkard pollen sampler, c – Pollen Monitor BAA500. Profile and bin numbers, layer heights, and lidar-derived optical values of selected layers for each station (mean values ± standard derivation of layer-mean values of all profiles) are given (PDR – particle linear depolarization ratio, BAE – backscatter-related Ångström exponent).**

The mean values of PDR and BAE in Table 2 are the averages of the layer-mean values (in the selected layers) of all selected profiles per each station. Averaged layer-mean values of PDRs in pollen layers of four stations are slightly enhanced (from about 0.04 to 0.09) than the background conditions, suggesting the presence of non-spherical particles in the atmosphere.
"

For the clarity, we show here the layer-mean values of each profile given by colour boxes in the following figure, but in the manuscript we kept the original figure showing the averaged ones from layer-mean values.

[Figure]

232 "Note that if there are many profiles, it is possible to use the mean values of pollen layers instead of each bin." Yes but what does it mean? Is this related to the squares reported in fig.2? The authors should dwell more on the description of their dataset and procedures of data processing.

Thank you for the comment, we agree that this sentence is a bit confusing, and therefore we decide to remove this sentence. The squares in fig.2 show the averaged values of layer-mean values of all selected profiles for each station.

236 What does "good dataset" mean?

We have changed the "good dataset" to "under ideal conditions" for the clarity.
"
Under ideal conditions (i.e. two aerosol populations present in the mixture, with different mixing ratio at different height or time), the unique solution can be found for the coefficients ($a_1$, $a_2$) with a high accuracy.
"

240 "the depolarization ratio of the background particles can be reasonably estimated or assumed" Please refer to what I considered before: to what extent the assumption on the depolarization ratio of the background particles on a given day can be used to infer the characteristic pollen depolarization? how the uncertainties on such assumption influence the result?

The assumption on the characteristic of background particles has no influence on the estimation of the characteristic of pollen. The characteristic values of 2 aerosol types can be estimated separately.
The depolarization ratio of the background particles can be reasonably estimated or assumed, thus the $BAE_{bg}$ can be calculated. For pollen, we assume the BAE of pure pollen ($BAE_{pollen}$), so as to calculate the pollen depolarization ratio.

If we take Leipzig as an example, the derived relationship of BAE and PDR is:
$$\left(\frac{355}{532}\right)^{-BAE} = \frac{a_1+a_2}{(PDR+1)} - a_2 \text{, with } a_1 = 1.9745, a_2 = 3.7540$$
Simple conversion yields:
$$BAE = -\frac{log\left(\frac{a_1+a_2}{(PDR+1)} - a_2\right)}{log\left(\frac{355}{532}\right)}$$
If there is only background aerosols: when PDR=0.01, BAE can be calculated as 1.61; if PDR = 0.02, BAE is calculated as 1.54; if PDR = 0.03, BAE is calculated as 1.46.
When we assume there is only pollen, for different assumed $BAE$ values, we can calculate different PDR. In the paper we give $DR_{pollen}$ range for the assumption of $BAE_{pollen}$ from -0.5 to 0.5.
We have added description in the section 2.4 for the clarity:
"
The relationships between simulated BAE and PDR under each assumption are shown in Fig. 1d: the bottom-right (top-left) boundary point is determined by the BAE and the depolarization ratio (DR) of the depolarizing (non-depolarizing) particles, shown as dark brown squares (light brown triangles); whereas the curve shape of fitting lines is determined by Eq. (7), i.e. different values of $a_1$ and $a_2$ defined by Eq. (8). Note that the two boundary points are independent, determined by the characteristic values (BAE and DR) of each aerosol type separately.
"

We agree with the referee that the assumption on the characteristic of background particles will affect to the pollen (or dust) backscatter coefficient retrieval (e.g. fig. 3e, fig. 5c, and fig. 6c). For different input values of depolarization ratio of background aerosols (the input $DR_{bg}$ was selected as 0.01, 0.02 or 0.03), different pollen (or dust) backscatters can be found as shown in the following figures for the selected cases in the manuscript. An underestimate of the $DR_{bg}$ will result in an overestimate of the pollen backscatter coefficient, thus a higher pollen backscatter contribution. Higher uncertainty on such an assumption (i.e. $DR_{bg}$) can be found when the depolarization ratios of considered depolarizing and non-depolarizing particles are of closer values, e.g. a larger discrepancy was found for HPB case1 than HPB case2, as the depolarization ratio of the depolarizing particle was 0.21 and 0.32, respectively.
We have added description in the revised version.
At the end of sect.3.3.1, we added:
"
The assumption on the depolarization ratio of the background particles ($DR_{bg}$) can affect the pollen backscatter coefficient retrieval. An underestimate of the $DR_{bg}$ will result in an overestimate of pollen backscatter coefficient.

For the given case example, if $DR_{bg}$ would be assumed as 0.01 instead of 0.03, a ~ 6 % higher pollen backscatter contribution (with a layer-mean value of 56 % instead of 51 %) would be obtained.

"

In sect.3.3.3, we added:

"

The layer-mean backscatter contribution of pollen (dust) for the selected case in period no.1 (no.2) was estimated as ~ 22 % (53 %), based on the evaluated pure depolarization ratios of 0.21 (0.32) and BAE of 0. If $DR_{bg}$ was assumed as 0.03 instead of 0.01, the layer-mean backscatter contribution of pollen (dust) for the selected case was estimated as ~ 11 % (49 %).

"

[Figure]

242 "the BAE of pure pollen can be assumed to be 0, as pollen grains are quite large particles" would this probably be the case if the pollen were transparent, but aren't they colored instead? Are the authors able to give an estimate of the angstrom coefficient of the pollens, from their dataset? This might be done by extrapolation, starting from the measured total angstrom coefficient and from the variability of the background aerosol relative contribution to it, given that it could be be assumed the value of the background aerosol angstrom coefficient (known from previous measurements?).

As discussed in our previous pollen paper (Shang et al. 2020), the extinction-related Ångström exponent (EAE) is characterized mainly by the particle size, whereas the backscatter-related Ångström exponent (BAE) depends on the particle size, shape, and the refractive index. In the case of pollen, we assume that the size is the crucial parameter. In ambient conditions, the size of pollen is more comparable with size of cloud particles than with aerosol particles. Thus we consider zero value reasonable. Furthermore, for large particles, we can assume a reasonable EAE of 0. The BAE and EAE are related by lidar ratios; our previous studies (Bohlmann et al. 2019; Shang et al. 2020) show no significant wavelength dependence on LR values (at 355 and 532 nm) for the measurements of the mixture of pollen and background aerosols.

We agree that the investigation of lidar ratio of pure pollen is important, and such pure pollen lidar ratios may show wavelength dependency for different pollen types, thus resulting a non-zero BAE. We are working on such scientific question which is, however, out of the scope of this manuscript.

We have added the descriptions and discussions in the revised version (section 2.4) for the clarity:

"

Here, we investigate, mathematically, the relationship of the backscatter-related Ångström exponent (BAE) and the particle linear depolarization ratio (PDR). Note that the BAE depends on the particle size, shape, and complex

refractive index (e.g. Miffre et al., 2020; Mishchenko et al., 2002), and thus demonstrate higher sensitivity to the changes in aerosol mixture composition.
"

Thank you for the suggestion about the extrapolation method. It is not accurate for such an extrapolation to estimate the pure Ångström exponent with the database used in this manuscript, regarding the overlap limitations. We have added descriptions of overlap information in the revised version (section 3.2):
"

The bottoms of the pollen layers are limited due to the overlap of the lidar instrument (the lowest reliable height after the quality control tests is about 900, 500, 700, or 600 m agl, for KUO, WAW, HPB, or LEI, respectively), whereas the tops are defined as the lowest observed layers based on the gradient method applied on both BSCs and PDRs.
"

However, we have other measurements with additional near-field telescope of other pollen campaigns which can provide optical properties down to ~150 m agl. We can apply the extrapolation method to estimate of the Ångström exponent. Also some laboratory measurements of fresh pollen in the in atmospheric conditions would be a good way to evaluate such values. This work is included in our future work plan. We have also added descriptions in the revise version (section 4):
"

However, the uncertainty on the assumed BAE of pure pollen will introduce non-negligible bias. If the true value of pollen BAE is between -0.5 and 0.5, relative uncertainties on estimated pollen depolarization ratios were found between 14–30 %. Thus, measuring the Ångström exponent of pure pollen, for example in laboratory experiments (in atmospheric conditions), would be beneficial and would certainly improve the determination of pure pollen depolarization ratios.
"

358 "This study shows that automatically retrieved lidar data profiles (using SCC) are suitable for pollen characterizations. The proposed methodology demonstrated a first step towards automated pollen detection in lidar networks." I hope I have brought the authors aware of possible problems related to their assumptions and their method, and have suggested to them how to expand the work to address these issues, which appear to me critical. I believe that the conclusion should be rewritten by better highlighting the limits of applicability of the method, which in my opinion, at least in the present state, can only provide indirect confirmation to the predictions of the dust and pollen models, rather than proposing itself as an independent method for the their detection and characterization.

Thank you for the comment. We have added the limitation in the conclusion:
"

This study shows that automatically retrieved lidar data profiles (using SCC) are suitable for pollen characterizations. The method was demonstrated for sites at which we have seldom or none (e.g. Warsaw and Kuopio) long-range-transported dust. Additional information, e.g. dust-free period from dust models or fluorescence information to identify dust and pollen (Veselovskii et al. 2021), is needed to exclude dust impact in the areas where dust is present. The proposed methodology demonstrated a first step towards automated pollen detection in lidar networks.
"

**Response to Dr. Alain Miffre**

Thank you for carefully reading the manuscript and providing useful suggestions to improve the paper. The replies to the comments are given below. The comments are in blue with our responses in black. The sentences in the manuscript are between the quotation marks, with the modifications in the revised manuscript in red.

Let me please thank ACP for giving me the opportunity to join the discussion. I indeed find the subject of pollen observations with lidar important and appreciated the performed field measurements which will interest several readers.

I however have concerns about the proposed data analysis as the retrieved depolarization of pure birch disagrees with the literature by Cao et al. (Cao, X., Roy, G. and Bernier, R.: Lidar polarization discrimination of bioaerosols, Opt. Eng., 49(11), 116201, doi:10.1117/1.3505877, 2010). I hope the questions below will help improving the analysis of these interesting lidar data and help future readers identifying the benefits / limitations of the proposed methodology:

- From Cao et al. (2010), we indeed learnt that the pure birch depolarization is equal to 32 +/- 2 % but the present algorithm arrives to 24 +/- 5 %. Can the authors explain this discrepancy? It may interest potential readers from the pollen community. I think future readers may then wonder about the need for developing an algorithm (with inherent assumptions) to retrieve the depolarization of pure birch while Cao et al. published its value: can the authors explain?

Thank you for the comments.
Cao et al. (2010) determined a linear depolarization ratio at 532 nm for paper birch of 0.33 ± 0.004 in the laboratory; Cholleton et al. (2020, European Lidar Conference – ELC, "Laboratory evaluation of the (UV, VIS) lidar depolarization ratio of pure pollen grains at exact backscattering angle") present a laboratory evaluation of PDR at 532 nm for birch pollen of 0.31 ± 0.06 using a polarimeter; whereas in Shang et al. (2020) we reported a linear depolarization ratio at 532 nm for ambient silver birch of 0.24 ± 0.01 under the assumption that the backscatter-related Ångström exponent between 355 and 532 nm for birch pollen was zero. As stated in Bohlmann et al. (2019) and Shang et al. (2020), it has to be kept in mind that these two experiments were conducted in quite different environments and conditions. Under ambient conditions the fresh pollen grains are more spherical than dried pollen grains (for laboratory measurements performed in an aerosol chamber), and therefore less depolarizing. Furthermore, these birch pollen are from different species. Examples of microphotographs of fresh and dry birch pollen grains are given in Fig. 1 below.

(a)                                         (b)

[Figure]

Figure 1. Microphotographs of birch pollen grains: (a) *Betula pendula* (fresh, rehydrated (water)), photographer: Halbritter, H., source: PalDat – a palynological database (https://www.paldat.org,). (b) Scanning electron microscopy image of laboratory birch pollen grain, University of Lyon, source: Cholleton et al. (2020, ELC).

The laboratory measurements of different types of depolarizing particles are not always available from literature, especially in atmospheric conditions (both the sample and environmental wise). The novel method present in the manuscript provides a way to estimate the relationship between measured PDR and BAE, and thus to evaluate the characteristic values (depolarization ratio or BAE) of the pure aerosol type if one parameter is known or can be reasonably assumed in real atmospheric conditions.

- The pure birch depolarization is retrieved "under the assumption that the BAE between 355 and 532 nm should be zero (± 0.5) for pure pollen". In the meanwhile, from light scattering theory (see Mishchenko et al., 2002 for example), we know that the value of BAE is a complex relationship between size, shape and chemical composition. Can the authors then explain their chosen assumptions: an identical BAE for birch and birch and grass, a zero value for BAE for pure pollen, a 0.5 uncertainty on BAE ?

Thank you for the comments.

In the manuscript, we provided an easy-to-apply algorithm to determine mathematically the relationship between PDR and BAE of aerosol mixture with two aerosol types. Based on the derived fitting curve, characteristic values of the couple of BAE and DR the pure aerosol type can be determined as stated in the manuscript:

"
The relationship between lidar-derived BAE and PDR of total particles is fixed for the mixture of two aerosol types.

…

Regarding the fitting Eq. (9), the value couple of $BAE_x$ and $DR_x$ of one pure particle type (*pollen* or *bg*) should be located on the fitting curve theoretically (or under ideal conditions). Thus, with the knowledge of one parameter, the other can be evaluated.
"

It is known that BAE depends on size, shape and composition, however, in the case of pollen, we assume that the size is the crucial parameter. In ambient conditions, the size of pollen is more comparable with size of cloud particles than with aerosol particles. Thus we consider zero value reasonable (e.g. for very large cirrus crystal particles such a value would be also zero).

The BAE and EAE (extinction-related Ångström exponent) are related by lidar ratios. Our previous studies (Shang et al. 2020) show no significant wavelength dependence on LR values for the measurements of the mixture of pollen and background aerosols. We agree that the investigation of lidar ratio of pure pollen is important, and such pure pollen lidar ratios may show wavelength dependency for different pollen types. We are working on such scientific question which is, however, out of the scope of this manuscript. We added the description of BAE in the revised version for the clarity:

"
Here, we investigate, mathematically, the relationship of the backscatter-related Ångström exponent (BAE) and the particle linear depolarization ratio (PDR). Note that the BAE depends on the particle size, shape, and complex refractive index (e.g. Miffre et al., 2020; Mishchenko et al., 2002), and thus demonstrate higher sensitivity to the changes in aerosol mixture composition.
"

Values of Ångström exponent of pure pollen are not available in the literature; there were some laboratory measurements of depolarization ratios of pure pollen grains, but no measurement of fresh pollen in the air is available. Hence, we estimate the depolarization ratio under some assumptions of BAE values. We added descriptions in the conclusion of the revised manuscript:

"
However, the uncertainty on the assumed BAE of pure pollen will introduce non-negligible bias. If the true value of pollen BAE is between -0.5 and 0.5, relative uncertainties on estimated pollen depolarization ratios were found between 14–30 %. Thus, measuring the Ångström exponent of pure pollen, for example in laboratory experiments (in atmospheric conditions), would be beneficial and would certainly improve the determination of pure pollen depolarization ratios.
"

*Supplement of*

**Pollen observations at four EARLINET stations during the ACTRIS-COVID-19 campaign**

Xiaoxia Shang et al.

*Correspondence to*: Xiaoxia Shang (xiaoxia.shang@fmi.fi)

The copyright of individual parts of the supplement might differ from the CC BY 4.0 License.

**1 Synthetic examples for 3 aerosol mixture**

The method presented in the manuscript can be applied if there are only two aerosol populations present in the mixture. Here we discussed two cases when there are 3 aerosol types present in the mixture, using synthetic examples. In these cases, we assume there are dust, pollen and non-depolarizing background aerosols in the aerosol mixture and chosen the input characteristic values shown in Table S1. The backscatter coefficients profiles (at 532 nm) of each type were simulated (Fig. S1–S2 a), as well as the total particle backscatter coefficient. Using these initial characteristic values, the particle linear depolarization ratio (PDR) at 532 nm and the backscatter-related Ångström exponent (BAE) between 355 and 532 nm of total particles were calculated (Fig. S1–S2 b–c). The scatter plot of all bins (black circles) in the simulated profiles is given in Fig. S1–S2 d.

**Table S1. Initial characteristic values of dust, pollen and non-depolarizing background aerosols, as input of the synthetic examples.**

| Aerosol type | Depolarization ratio at 532 nm ($\delta$) | Backscatter-related Ångström exponent 355-532 (BAE) | Half width (Gauss.) | Layer centre case 1 (Fig. S1) | Layer centre case 2 (Fig. S2) |
|---|---|---|---|---|---|
| Dust | 0.40 | 0 | 1 km | 2.5 km | 1.5 km |
| Pollen | 0.30 | 0 | 1 km | 1.5 km | 1.5 km |
| Background | 0.01 | 2 | 3 km | 2.0 km | 2.0 km |

[Figure]

(a)  (b)  (c)  (d)

**Figure S1. Case 1: (a) Synthetic vertical profiles of the backscatter coefficient (BSC) at 532 nm of dust, pollen, background and total particles. Synthetic profiles of (b) the particle linear depolarization ratio (PDR) at 532 nm, and (c) the backscatter-related Ångström exponent (BAE) between 355 and 532 nm, of total particles, under initial values given in Table S1 – case 1. Two layers are defined (layer 1 – orange, layer 2 – green) in dashed boxes (a–c). (d) Scatter plot of PDR and BAE for the synthetic case. The dashed fitting lines are determined by Eq. (7), for layer 1 (orange), layer 2 (green), and all bins (black). Colour dots (inside each layer) and black circles (entire profile) present each bin.**

*Case 1:*

We defined 2 layers here (orange and green dashed boxes in Fig. S1 a–c): layer 1, from 3.0 to 3.8 km, where there are only dust and background aerosols present; layer 2, from 0.2 to 1.0 km, where there are only pollen and background aerosols present. As in both layers, there are only two aerosol types (one depolarizing and the other non-depolarizing), perfect fitting lines can be found by applying the method, i.e. Eq. (7) in the manuscript (orange and green dashed lines in Fig. S1 d). However, bins outside these 2 layers (black circles without colour) do not fit to either fitting line, but locating between them or on the top-left edge (where there are only background aerosols). When applying the method to all bins, a fitting line (in dashed black) was found between the ones of layer 1 and layer 2. As a result, under the assumptions that there are two aerosol types in the given aerosol mixture and that the BAE of this "depolarizing aerosol" should be zero, the pure depolarization ratio of this "depolarizing aerosol" was estimated as 0.36, which is a value between the initial depolarization ratios of pure dust (0.4) and pure pollen (0.3).

[Figure]

**Figure S2. Similar as Fig. S1, but for Case 2 (Table S1).**

*Case 2:*

In this case, the initial layer heights and layer depths of dust and pollen aerosol were selected to be the same (Table S1, Fig. S2 a). As these two depolarizing aerosols are well mixed with homogenous mixing ratio in this case, a perfect fitting line can be found for the PDR and BAE of total particles (black dashed line in Fig. S2 d) with all bins located on the fitting line, resulting a characteristic depolarization ratio of 0.36 under the assumption of characteristic BAE of zero. The theoretical fitting lines (from equations using initial values of Table S1) for dust and background aerosol mixture or pollen and background aerosol mixture are also given as orange or green dashed lines.

Not that the cases in the reality can be more complicated than the considered examples.

For the given examples, we can conclude that for an aerosol mixture of two types of depolarizing aerosols and one non-depolarizing aerosol, the estimated depolarization ratio for "depolarizing aerosols" represent the combination of two present depolarizing aerosols, with a value between the characteristic (pure) depolarization ratios of each type. For aerosol mixture with more types, the estimated depolarization ratio for "depolarizing aerosols" represent a combination of all present depolarizing aerosols, with a value inside the range of the characteristic (pure) depolarization ratio values of each aerosol type.

However, authors recommend using the method under the constraints mentioned in the manuscript.

**2 Dust concentration profiles from the NMMB/BSC-Dust model**

Images were downloaded from https://ess.bsc.es/bsc-dust-daily-forecast, with data from the NMMB/BSC-Dust model, operated by the Barcelona Supercomputing Center.

HPB Period no.1: 7–8 May 2020

[Figure]

HPB Period no.2: 18 May 2020

[Figure]

KUO 23–26 May 2020

[Figure]

NMMB/BSC-Dust
Kuopio: 62.74N, 27.54E
Dust Forecast at 12 UTC Sun, 24 May 2020

NMMB/BSC-Dust
Kuopio: 62.74N, 27.54E
Dust Forecast at 18 UTC Sun, 24 May 2020

NMMB/BSC-Dust
Kuopio: 62.74N, 27.54E
Dust Forecast at 00 UTC Mon, 25 May 2020

NMMB/BSC-Dust
Kuopio: 62.74N, 27.54E
Dust Forecast at 06 UTC Mon, 25 May 2020

NMMB/BSC-Dust
Kuopio: 62.74N, 27.54E
Dust Forecast at 12 UTC Mon, 25 May 2020

NMMB/BSC-Dust
Kuopio: 62.74N, 27.54E
Dust Forecast at 18 UTC Mon, 25 May 2020

NMMB/BSC-Dust
Leipzig: 51.35N, 12.43E
Dust Forecast at 00 UTC Tue, 26 May 2020

NMMB/BSC-Dust
Leipzig: 51.35N, 12.43E
Dust Forecast at 12 UTC Tue, 26 May 2020

NMMB/BSC-Dust
Leipzig: 51.35N, 12.43E
Dust Forecast at 18 UTC Tue, 26 May 2020

LEI 26–27, 30–31 May 2020

[Figure]

NMMB/BSC-Dust
Leipzig: 51.35N, 12.43E
Dust Forecast at 18 UTC Tue, 26 May 2020

NMMB/BSC-Dust
Leipzig: 51.35N, 12.43E
Dust Forecast at 00 UTC Wed, 27 May 2020

NMMB/BSC-Dust
Leipzig: 51.35N, 12.43E
Dust Forecast at 12 UTC Sat, 30 May 2020

[Figure]

WAW Period no.1: 26–29 May 2020

[Figure]

[Figure]

WAW Period no.2: 31 May 2020

[Figure]

*Acknowledgements.* The authors acknowledge the data and/or images from the NMMB/BSC-Dust model, operated by the Barcelona Supercomputing Center (http://www.bsc.es/ess/bsc-dust-daily-forecast, last access: 17 Dec 2021).